# Knowledge, Attitude, and Behavior toward COVID-19 Vaccination in Young Italians

**DOI:** 10.3390/vaccines11010183

**Published:** 2023-01-15

**Authors:** Shizuka Kibi, David Shaholli, Vanessa India Barletta, Francesca Vezza, Marcello Gelardini, Carla Ardizzone, Daniele Grassucci, Giuseppe La Torre

**Affiliations:** 1Department of Public Health and Infectious Diseases, Sapienza University of Rome, Piazzale Aldo Moro 5, 00185 Rome, Italy; 2Skuola.net, Via Arrigo D’Avila 37/G, 00179 Rome, Italy

**Keywords:** knowledge, attitude, behavior, COVID-19 vaccination, young Italians

## Abstract

*Purpose*: The knowledge, attitudes, and behavior of young Italians towards the COVID-19 vaccination were analyzed in order to provide information useful to elaborate the strategies that can be implemented to obtain the best possible vaccination coverage in this population. *Methods*: A cross-sectional study was conducted on 5313 young people aged between 11 and 30 years. Data were collected through an online survey during the period from 1 to 10 March 2021. The answers to the questionnaire were analyzed using SPSS statistical software. Intention to vaccinate was studied by univariate analysis using Pearson’s chi-square test to assess differences between groups for categorical variables, and by multivariate analysis applying the binary logistic regression model, and the Hosmer–Lemeshow test was performed to assess goodness of fit. *Results*: Television (32.1%), internet/search engine (25.9%), and social networks (10.4%) were the main sources of information for young Italians. The survey analysis showed that 74.5% (3956) of the respondents were willing to be vaccinated against COVID-19 versus 25.5% (1357) who were against it. Demographic data, in particular, age, gender, experience with influenza vaccine, and level of knowledge about the disease, were significant determinants (*p* < 0.001) for the choice to vaccinate against COVID-19. *Discussion*: These results suggest that in order to implement the vaccination campaign, correct information is needed to improve awareness of the vaccine and COVID-19, while also taking into account the target group, which differs not only in age but also in the sources of information used compared to the adult population.

## 1. Introduction

COVID-19 has had a huge impact on the world’s population; social isolation due to public health measures, such as quarantine, resulted in discomfort that was especially reflected in schools, where distance learning caused students symptoms of stress, interaction problems, and lack of motivation to study [1].

During this pandemic, the anti-COVID-19 vaccination proved to be a very effective prophylactic measure in preventing infection and especially severe disease [2].

On 22 December 2020, marketing authorization for the Comirnaty vaccine for people aged 16 years and older was given in Italy by the Italian Medicines Agency (AIFA) [3]. Comirnaty was the first vaccine to be approved in the European Union for this age group. Following a recommendation by the Committee for Medicinal Products for Human Use (CHMP) of the European Medicines Agency (EMA), on 28 May 2021, marketing authorization for the Comirnaty vaccine was extended in the European Union to children aged 12 to 15 years. On 25 November 2021, the EMA recommended the extension of the vaccine indication to children aged 5 to 11 years [4], followed by authorization by the AIFA (1 December 2021). On 29 January 2021, the CHMP gave a positive opinion regarding the use of another COVID-19 vaccine, ChAdOx1-S, indicated for active immunization to prevent COVID 19 caused by SARS-CoV-2 in individuals 18 years of age and older. Up until now, however, ChAdOx1-S is not currently authorized for use in children [5].

On 7 June 2022, the FDA approved emergency use of the Comirnaty vaccination in children aged 6 months to 4 years, with the dose reduced to 3 µg, divided into three administrations. On 19 October 2022, the European Union and CHMP also approved the extension of the use of the Comirnaty vaccine to children aged 6 months to 4 years.

Referring to the Italian distribution of reported infection, we observed that the percentage of diagnosis among people between 10 and 19 years old (11.4%) was similar to the percentage recorded for adults from 40 to 49 years old (15.5%) [6]; this could be related to teenager vaccination coverage (83.92%), which was lower compared to adults (90.50% in the 50- to 59-year-old age group) [7]. These data led us to investigate the vaccination adherence of young people, taking into consideration not only the importance of protecting the most at-risk groups by limiting cases of infection, but also the effects of COVID-19 on this population, which, even if less frequently, can present serious complications, such as multisystem inflammatory syndrome [8]. Not surprisingly, WHO has listed vaccine hesitancy as one of the top ten public health threats worldwide [9].

Several studies around the world have found that factors influencing teenagers’ willingness to vaccinate can be attributed primarily to confidence in the vaccine’s efficacy and safety.

Analysis of a survey conducted in China reported that young people who had not previously undergone flu vaccination tended not to vaccinate against COVID-19 or were hesitant. Individuals who did not perceive a risk of becoming infected also had greater resistance or hesitation to be vaccinated, while this decreased in young people who thought the vaccine was more effective [10]. Thus, having greater confidence in the vaccine promotes anti-COVID-19 vaccine adherence. In addition, there is also the perception of risk for the disease, whereby knowing someone diagnosed with COVID-19 is associated with intention to vaccinate [11].

Other studies have emphasized the influence of age on intention to vaccinate. In a group of Dutch youth (12–18 years old), intention to be vaccinated was related to age, perceived personal benefit (protecting one’s health), social benefits (getting rid of restrictions), and vaccination adherence by parents and peers. Older adolescents showed greater willingness to vaccinate (16–17 years (80%) vs. 12–15 years (68%)), as did those with higher levels of education (78% vs. 64%) [12].

According to Zhang et al. 2022, however, the trend in vaccination adherence as age increases is then reversed, and vaccination hesitation has a higher prevalence in young adults between 18 and 21 than in older adolescents (16/17 years). Reasons for hesitation include underestimating the risks of infection, thinking that protection from innate immunity is sufficient, special physical conditions not suitable for vaccination, and conspiracy theories about vaccination, which are more common among young adults than older teenagers [13]. Cai et al. 2021 reported that younger adolescents were more likely to vaccinate than older adolescents. The reasons for not vaccinating were, mainly, unknown long-term side effects [14].

A study carried out by Middleman et al. [15] showed that, before the availability of COVID-19 vaccine for adolescents (waves 1 and 2), the main reasons to get the vaccine were the intention to protect themselves (49–53%), to protect everyone in the family (47–52%), the fact that vaccination is the best way to avoid a potentially serious disease (37–38%), and the willingness to be safe around other people (38–39%).

Another study carried out in Lebanon in Spring 2020, when a COVID-19 vaccine was not available, indicated a high level of optimism among young people about the possibility of finding a vaccine for COVID-19 (88.2%), as well as readiness to take the vaccine when available (66.2%) [16].

These data suggest thinking about the importance of perceived long-term side effects among this population. Despite high COVID-19 vaccine awareness, teenagers were afraid of the potentially severe, even if rare, side effects. The study identified concern about vaccine efficacy and the potential long-term health impact as the main obstacles to vaccination. The main sources of information reported were the family, mainstream media, and, to a lesser extent, teachers and schools [17].

The aim of this study was to understand the knowledge, attitudes, and behaviors of young Italians toward anti-COVID-19 vaccination in order to provide useful information to evaluate strategies that can be implemented to achieve the best possible vaccination coverage in this population.

## 2. Materials and Methods

A cross-sectional study was carried out on data from an online survey with the collaboration of “Skuola.net”, an Italian website for information and insights for secondary school, high school, and university students, and which was already involved in other surveys [18,19,20,21].

The survey was open to all users, no registration required, so there were no limits to inclusion, and participation was free and anonymous. Participants filled out an online questionnaire on the Skuola.net website (a site dedicated to school and university students with more than 6,000,000 unique users/month), during the period between 1 and 10 March 2021. A total of 5313 youth aged 11 to 30 responded to questions regarding COVID-19 and the COVID-19 vaccine. The highest prevalence subgroup, 59.8% (3175), was 15 to 18 years of age, followed by the 11 to 14 years age group at 22.0% (1168).

The questionnaire included 18 multiple choice questions: one group of questions concerning the demographic data of the respondents, such as gender, age, geographical area of residence, and level of education, and one group aimed at investigating knowledge, attitudes, and behaviors (KAB: Knowledge, Attitudes, Behaviors) of the participants towards anti-COVID-19 vaccination. This included the use of descriptive and inferential statistics. Percentages and frequency scores for respondent characteristics are presented in Table 1, and attitudes and knowledge are presented in Table 2. A pilot study was carried out to test the reliability of the questionnaire and it showed a clear indication of good psychometric properties (Cronbach’s alpha = 0.794).

A univariate analysis, including chi-square for categorical variables, was performed to assess the differences among groups of descriptive variables associated with willingness to be vaccinated against COVID-19. A multiple logistic regression analysis was conducted, taking into consideration the willingness to be vaccinated against COVID-19 as a dependent variable. Univariate analyses were considered to test the inclusion of each explanatory variable in the final models. Effects with a *p*-value < 0.20 were included in the multivariate analysis model. Results of the multiple logistic models were presented as odds ratios with 95% confidence interval (95% CI). The quality of the model’s fit was assessed by the Hosmer–Lemeshow test.

All analyses were conducted using SPSS for Windows 27.0 (IBM, Armonk, NY, USA). Statistical significance was set at *p* < 0.05.

### Ethical Issues

The study was conducted in accordance with the Declaration of Helsinki and approved by the Institutional Review Board (Sapienza University of Rome—Policlinico Umberto I Ethical Committee n.109/2020). Informed consent was obtained using an ad hoc consent form in the presentation of the survey. In the case of minors, guardians provided informed consent.

## 3. Results

### 3.1. Characteristics of the Respondents

The sample consisted of 5313 subjects: 3390 (63.8%) were women vs. 1689 (31.8%) men; furthermore, there was greater participation by the young (11–14 years old (1168 (22.0%)); 15–18 years old (3175 (59.8%))). Consistent with the percentages of the age groups, the most frequently earned qualification was the middle school diploma (3115 (58.6%)), while the greatest involvement was recorded among young people from the regions of northern Italy (2344 (44.1%)) (Table 1).

### 3.2. Attitudes and Knowledge

Interestingly, television (1708 (32.1%)) and internet/search engines (1377 (25.9%)), followed by social networks (550 (10.4%)), were the main sources of information for young Italians, whereas only very few people reported being informed by a physician.

A total of 11.4% (604) declared that they had never been vaccinated, and 31.5% (1672) stated that they had not been vaccinated against the flu that same year due to lack of availability, while 38.9% (2067) said they had not because they did not consider it useful; this proves the weakness of the anti-flu vaccination campaign among the young population.

Regarding COVID-19 vaccination timing, most young people expressed willingness to take this treatment “only after all the health care workers and vulnerable people get the vaccine” (1895 (35.7%)), although we observed alongside this a large percentage who answered “as soon as possible” (1412 (26.6%)). The answers to the question “are you going to get the COVID-19 vaccine?” (qualitative variable) were converted to a quantitative variable before analysis; we considered as 1 (“yes”) the answers “basically yes, but I still have some doubts” and “definitely yes”, and as 0 (“no”) the other answers “absolutely not”, “I don’t know”, and “basically no, I still have too many doubts”.

According to this division, 74.5% (3956) of the respondents intended to be vaccinated against COVID-19 against 25.5% (1357) who did not want to be vaccinated (Table 2).

### 3.3. Univariate Analysis

In univariate analysis, willingness to be vaccinated against COVID-19 (“yes” or “no”) was seen as a dependent variable, and demographic data as independent variables. The resultant % values were all found to be statistically significant (*p* < 0.001) using the Pearson chi-square test (significance level: *p* < 0.05). The percentage of female subjects (2594 (76.5%)) who intended to be vaccinated was slightly higher than that of males (1230 (72.8%)), whereas, based on age, the greatest willingness was found between 15 and 18 years (2585 (81.4%)), followed by the groups 19–22 years (450 (77.7%)) and 11–14 years (796 (68.2%)).

As regards the level of education, COVID-19 vaccination was more successful among those with a middle school diploma (2437 (78.2%)) or a high school diploma (997 (80.9%)), while, paradoxically, the percentages were lower among those who had graduated (106 (43.3%) or obtained a postgraduate degree (45 (24.9%)). In southern Italy, we saw a greater intention to be vaccinated, particularly relative to the north (Table 3).

### 3.4. Multivariate Analysis with Dependent Variable of Willingness to Be Vaccinated against COVID-19

In the multivariate analysis, we considered as a dependent variable the willingness to be vaccinated against COVID-19. In this analysis of binary logistic regression, we also calculated the “knowledge score” by assigning a score of 1 for the only correct answer for each of the three questions: “what is SARS-CoV-2?” (“a virus” = 1), “what is the disease associated with SARS-CoV-2?” (“COVID-19” = 1), and “COVID-19 is the correct answer: but what is it more exactly?” (“a disease associated with SARS-CoV-2” = 1), and a score of 0 for the other answers. Results were reported in terms of an odds ratio (OR) and associated 95% confidence intervals (95%CI).

The sample was stratified by gender (women showed higher odds of getting vaccinated compared to men) (OR = 1.22, 95%CI, 1.06–1.40)), and age (greater than or equal to, or under, the age of 18), and persons of legal age (OR = 0.46, 95%CI, 0.40–0.54) had less intention to be vaccinated than the younger ones. It must also be emphasized that the willingness increased in those who were already familiar with the vaccination, having received it at least once (OR = 2.78, 95%CI, 2.35–3.27) or in those who had received the flu vaccine in 2020 (OR = 1.24, 95%CI, 0.86–1.78). Most importantly, increasing knowledge scores were associated with higher odds for the willingness to be vaccinated (OR = 1.89, 3.10, and 3.61 for scores of 1, 2, and 3, respectively) (Table 4).

## 4. Discussion

Given the importance of high vaccination coverage even among young people in dealing with this pandemic, some studies have already been carried out that have investigated the relationship between socio-demographic conditions, the level of knowledge about COVID-19, and the willingness to be vaccinated.

After AIFA approval (31 May 2021) of the indication of the COVID-19 vaccine also for the 12–15 age group [22], in Italy since 3 June 2021 vaccination has been extended to the entire population from 12 years old. Vaccine coverage in our country among young people, as of 2 November 2022, reached 83.9% for the 12–19 age group and 91.5% for the 20–29 age group [6]. The gap between these numbers and the vaccination intent recorded in the March 2021 survey (74.5% (3956)) may be due in part to preventative measures taken by the government. In fact, with the announcement of the mandatory “COVID-19 green certification” (green pass) for various social activities (shows, cinemas, spas, swimming pools, gyms, and indoor restaurants) by Prime Minister Mario Draghi on 22 July 2021, followed by the DECREE-LAW 6 August 2021, n. 111, an increase of 24.5% in the 12–19 age group was recorded between 27 July and 2 August 2021 [23].

Compared to the results of the present study, similar percentages for the intention to be vaccinated were also found in the USA (75.9%), in a survey conducted when the COVID-19 vaccine had still not been approved (30 October 2020) in which young people aged between 14 and 24 answered [24]; in China, where middle and high school students answered the questionnaire between 27 November 2020, and 12 March 2021, and a high percentage of acceptance (75.59%) of the COVID-19 vaccine for the future was recorded [95% (CI) 73.00–78.18%] (age of participants: 12–20 years); in Manitoba, Canada, during November to December 2020 (65.4%) [25]; and in the Netherlands (73%, survey conducted between 22 June 2021 and 27 July 2021, age: 12–18 years old) [12].

Another study in China found a lower percentage (60.0% (5497)) [10], and there was also a little less willingness in Hong Kong, where out of 2609 adolescents who filled out the questionnaire by 30 June 2021, only 1007 (39%) intended to be vaccinated against COVID-19 [11].

Since our survey had voluntary participation, we observed a greater interest among the girls (3390 (63.8%)) than the boys (1689 (31.8%)) in answering the questionnaire, and in contrast with some studies that have found no significant differences between the two sexes in the intention to be vaccinated [11,12,25], we saw a greater willingness in the female gender in Italy (OR = 1.22, CI95%, 1.06–1.40).

The age variable has led to diverse outcomes in the various publications. It is interesting to note that respondents in the age group that can make its own decision showed a lower intention to get the COVID-19 vaccination. In fact, this study showed that individuals who are of legal age have lower odds of getting COVID-19 vaccine (OR = 0.46, 95% CI, 0.40–0.54).

The vaccination acceptance of younger people has also been reported in China (multiple logistic regression analysis: *p* = 0.003, OR = 0.93, 95% CI: 0.89–0.98) [14], where Zhang et al. 2022 reported a higher prevalence of vaccine hesitancy among young adults aged 18 to 21 years compared to older adolescents 16 to 17 years (16.5% vs. 7.9%, *p* < 0.001) [13]. However, other studies have recorded no significant differences [9,10,18], while a trend that goes in the opposite direction as regards age was found in the Netherlands (80% 16–17 years vs. 68% 12–15 years), in line with a higher level of education, which implies a greater inclination to vaccination compared with a low level of education (78% vs. 64%) [12]. This latest data differs from what emerges from our analysis. The students with the highest educational level, such as a degree (106 (43.3%)) or postgraduate qualification (45 (24.9%)), have shown a lower intention to be vaccinated, an element that requires further study.

Regarding the behavior of the young population, it is important to take into account previous experience with flu vaccination, which increases confidence in the vaccine, as reported also in other studies with a particular focus on the vaccination status for the year 2020, prior to the start of the COVID-19 vaccination campaign [10,11]. In the case of vaccination of a minor, applying the general rules established in international charters (i.e., the 1989 New York Convention on the rights of the child and the 1997 Strasbourg Convention on the rights of the child), from the age of 12 the minor must in any case be heard. This right also needs to be applied to minors who have the ability to discern, in the procedures that concern them, in order to arrive at identifying the most suitable measure in accordance with the principle of the “*best interest of the child*” [25].

There is evidence that parents’ decisions and attitudes are directly related to the immunization status of adolescents [26]. However, in the case of a minor who would get the vaccination but one or both parents do not agree, the orientation of jurisprudence is that a judge can temporarily “suspend” the ability of a parent opposed to the vaccine. In this sense, the Italian National Bioethics Committee has stated that the minor needs to be heard by the pediatrician, and her/his will must prevail, “as it coincides with the best interest of his (her) psycho-physical health and public health” [27].

In Singapore, as of 10 February 2022, 96% of young people between 12 and 19 had been vaccinated with the first dose, a result also due to the implementation of a vaccination campaign started in March 2021 using material dedicated to this specific demographic group. The use of digital platforms like TikTok, Twitter, and Instagram contributed to providing accurate information with more effective means of communication [17].

The main sources of information indicated in Italy were television (1708 (32.1%), internet/search engines (1377 (25.9%)), and social networks (550 (10.4%)), so we could consider a type of communication adapted to the most used media, taking into account the need for specific and understandable content for this target group; the elements of greatest concern are safety, side effects, vaccine efficacy [28].

As evidenced by the knowledge score analysis, participants’ knowledge of SARS-CoV-2 is a determining factor in their intention to be vaccinated; in fact, other studies have highlighted that a better knowledge of the COVID-19 vaccine (OR = 1.96; 95% CI = 1.57–2.45) can have a positive impact on vaccine acceptance among young people [29]. Not understanding the extent of the COVID-19 pandemic led to vaccine hesitancy or refusal [10].

However, it must be said that, according to the study in Singapore for which the main barriers to vaccination were long-term risks and perceived efficacy, there is a gap between knowledge and its application; concerns about vaccine efficacy are in contrast with good awareness regarding the effectiveness of vaccination [17].

## 5. Strengths

There are no additional studies in the literature on attitudes toward COVID-19 vaccination among youth under the age of 18 in Italy.

Thanks to the website skuola.net, we have a quantitative sample, large (5313), free, and from all over Italy.

Selection criteria based on open and anonymous participation may have included topics that made people more motivated to provide complete and truthful responses.

The study found a significant correlation between demographics, COVID-19 awareness, and the intent to be vaccinated among young Italians.

## 6. Limitations

Our results do not reflect current vaccine uptake, as the survey was conducted in March 2021, before the vaccine indication was extended to children aged 12 to 15 years.

First of all, one methodological concern is the truthfulness of answers, and the risk of biases in population-based surveys, such as information and recall bias, cannot be excluded.

Secondly, participants were only students who visited the “Skuola.net” website, so a selection bias cannot be ruled out. Since the recruitment method was based on voluntary participation, the sample may not be representative of the population. However, previous surveys performed with skuola.net have shown results similar to those carried out with sample-based surveys.

Finally, in this survey, family socioeconomic status was not assessed, and it is possible that some factors, such as the educational level of parents, could have biased the results [30,31,32].

## 7. Conclusions

Young Italians have participated in the COVID-19 vaccination campaign beyond the percentage of willingness recorded in the survey, but given the impact that the pandemic has had on the world population, it is important for the future to understand which factors we can influence to achieve maximum vaccination coverage. The results of this study highlight the possibility of encouraging vaccination in young people by improving awareness of COVID-19 and COVID-19 vaccination. The factors that affect vaccination acceptance can be influenced by correct, specific, and adequate information for this target group, which must take into account age, level of education, and the most sought-after means of communication. Further research is needed to assess the need to distinguish subgroups by age and to better understand the relationship to education level. Implementing the flu vaccination campaign can also have a positive effect on the intent to receive the COVID-19 vaccine.

## Figures and Tables

**Table 1 vaccines-11-00183-t001:** Characteristics of the respondents.

Variable	*n*	%
*What is your gender?*		
Female	3390	63.8%
Male	1689	31.8%
Other/I’d rather not answer	234	4.4%
*Age (years)*		
11–14	1168	22.0%
15–18	3175	59.8%
19–22	579	10.9%
23–26	241	4.5%
27–30	150	2.8%
*Highest degree or level of school*		
Non-response	47	0.9%
Other	313	5.9%
Primary school diploma	180	3.4%
Middle school diploma	3115	58.6%
High school diploma	1232	23.2%
Degree	245	4.6%
Post-graduate degree (master’s degree, doctorate, specialization, etc.)	181	3.4%
*Macro region*		
North	2344	44.1%
Center	1319	24.8%
South and islands	1650	31.1%

**Table 2 vaccines-11-00183-t002:** Attitudes and knowledge.

Variable	*n*	%
*What is SARS-CoV-2?*		
I have no idea	458	8.6%
A bacterium	205	3.9%
A parasite	44	0.8%
A symptom	32	0.6%
A virus	4452	83.8%
A disease	122	2.3%
*What is the disease associated with SARS-CoV-2?*		
Coronavirus	785	14.8%
COVID-19	3623	68.2%
Flu	454	8.5%
Tumor	188	3.5%
Other	263	5.0%
*COVID-19 is the correct answer: but what is it more exactly?*		
Other	239	4.5%
A coronavirus strain	2201	41.4%
A parasite	229	4.3%
A common cold	166	3.1%
A disease associated with SARS-CoV-2	2478	46.6%
*How much did you keep informed on SARS-CoV-2 or COVID-19 in the last year?*		
Neither too much nor too little, a middle ground	2160	40.7%
Not at all	209	3.9%
A little	352	6.6%
More than a lot	836	15.7%
A lot	1756	33.1%
*What was your main source of information?*		
Friends/relatives	272	5.1%
Newspapers	398	7.5%
Influencer	60	1.1%
Internet/search engines	1377	25.9%
Books	118	2.2%
Doctor	211	4.0%
Themed websites	410	7.7%
Social network	550	10.4%
Television	1708	32.1%
Non-response	209	3.9%
*Have you ever been vaccinated in the past?*		
No	604	11.4%
I don’t know	167	3.1%
Yes	4542	85.5%
*Which vaccines did you get?*		
I don’t know/I don’t remember	305	5.7%
Mandatory ones and recommended ones (flu vaccine, for example)	2704	50.9%
Only non-mandatory ones (flu vaccine, for example)	94	1.8%
Only mandatory ones	1439	27.1%
Non-response	771	14.5%
*Did you get the flu vaccine this year?*		
No, I didn’t. I wanted to but I couldn’t do it due to lack of availability	1672	31.5%
No, I didn’t. I’m scared of side effects	425	8.0%
No, I didn’t find it useful	2067	38.9%
Yes, I did, as I do every year	782	14.7%
Yes, I did it just for this year because of the pandemic	199	3.7%
Non-response	168	3.2%
*If the vaccine weren’t free, how much would you be willing to pay to get a COVID-19 vaccine?*		
Even more than €100	177	3.3%
Up to €50	591	11.1%
From €50 to €100	362	6.8%
The maximum amount established by the national health service	588	11.1%
I cannot answer this question	2512	47.3%
I am not willing to pay	524	9.9%
Any price, I’d like to get vaccine as soon as possible	559	10.5%
*When would you want to get a COVID-19 vaccine?*		
As soon as possible	1412	26.6%
I cannot answer this question	192	3.6%
When there will be more scientific publications	202	3.8%
Only after all the health care workers and vulnerable people get vaccine	1895	35.7%
Not before one year	66	1.2%
In a few months	189	3.6%
Non-response	1357	25.5%

**Table 3 vaccines-11-00183-t003:** Univariate analysis with dependent variable of willingness to be vaccinated against COVID-19.

Variable	No n. (%)	Yes n. (%)	*p*
*Gender*			
Females	796 (23.5%)	2594 (76.5%)	
Males	459 (27.2%)	1230 (72.8%)	<0.001
Not answered	102 (43.6%)	132 (56.4%)	
*Age (years)*			
11–14	372 (31.8%)	796 (68.2%)	<0.001
15–18	590 (18.6%)	2585 (81.4%)	
19–22	129 (22.3%)	450 (77.7%)	
23–26	168 (69.7%)	73 (30.3%)	
27–30	98 (65.3%)	52 (34.7%)	
*Highest degree or level of school*			
Non-response	19 (40.4%)	28 (59.6%)	<0.001
Other	101 (32.3%)	212 (67.7%)	
Primary school diploma	49 (27.2%)	131 (72.8%)	
Middle School diploma	678 (21.8%)	2437 (78.2%)	
High school diploma	235 (19.1%)	997 (80.9%)	
Degree	139 (56.7%)	106 (43.3%)	
Post graduate degree (master’s degree, doctorate, specialization, etc.)	136 (75.1%)	45 (24.9%)	
*Macro region*			
North	667 (28.5%)	1677 (71.5%)	<0.001
Center	354 (26.8%)	965 (73.2%)	
South and islands	336 (20.4%)	1314 (79.6%)	

**Table 4 vaccines-11-00183-t004:** Multivariate analysis with dependent variable of willingness to be vaccinated against COVID-19.

Variable	OR (95% CI)
*Gender*	
Females	1.22 (1.06–1.40)
Males (reference)	1
Not answered	0.53 (0.39–0.71)
*Age group*	
Under 18 (reference)	1
≥18	0.46 (0.40–0.54)
*Macro region*	
North (reference)	1
Center	1.15 (0.98–1.35)
South	1.48 (1.26–1.72)
*Have you been vaccinated in the past*	
No (reference)	1
Yes	2.78 (2.35–3.27)
*Vaccination against flu in 2020*	
No (reference)	1
Yes	1.24 (0.86–1.78)
*Knowledge score*	
0 (reference)	1
1	1.89 (1.45–2.48)
2	3.10 (2.40–3.99)
3	3.61 (2.76–4–71)

## Data Availability

Data supporting the reported results can be requested from the authors.

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
