# Peer review of "Knowledge, Attitude, and Behavior toward COVID-19 Vaccination in Young Italians"

_vaccines, 2023, doi:10.3390/vaccines11010183_

Round 1

Reviewer 1 Report

The authors are presenting a survey study on COVID 19 vaccination knowledge, attitude and behavior among Italian minors and young adults.

I would like to share the following comments with the authors:

The authors introduce the paper with a focus on teenagers. First of all, I would recommend to review other studies from other countries more e.g., Middleman et al 2021 in Vaccines.

Second, their sampling is not matching this focus. Then there should be an age cap at 18 or maybe 21.

Third, for teenagers below 16, in most countries, parents or guardians are in charge of vaccination decisions. Thus, here the knowledge, attitude, behaviors of the teenagers <16 are less relevant. The authors should discuss this adequately, and potentially stratify their sample into an age-group that can take own decisions vs. those that cannot.

The study does not mention ethics approval, especially since there are minors involved. This requires parental consent. I do not see how this was obtained in a survey. If parental consent is missing, this study severely violates ethics regulations. The authors need to clarify this

Author Response

Reviewer 1

The authors are presenting a survey study on COVID 19 vaccination knowledge, attitude and behavior among Italian minors and young adults.

I would like to share the following comments with the authors:

The authors introduce the paper with a focus on teenagers. First of all, I would recommend to review other studies from other countries more e.g., Middleman et al 2021 in Vaccines.

Answer: many thanks for this suggestion. We added the following sentences:

A study carried out by Middleman et al [Middleman AB, Klein J, Quinn J. Vaccine Hesitancy in the Time of COVID-19: Attitudes and Intentions of Teens and Parents Regarding the COVID-19 Vaccine. Vaccines (Basel). 2021 Dec 21;10(1):4] showed that before the availability of SARS-CoV-2 vaccine for adolescents (wave 1 and 2) the main reasons to get the vaccine were the intention to protect themselves (49-53%), to protect everyone in the family (47-52%), the fact that vaccination is the best way to avoid a potentially serious disease (37-38%), and the willingness to be safe around other people (38-39%).

Another study carried out in Lebanon in Spring 2020, when SARS-CoV-2 vaccine was not available, indicated a high level of optimism among young people about the possibility to find a vaccine to Covid-19 (88.2%) and the readiness to take the vaccine, if available (66.2%)[ Sakr S, Ghaddar A, Sheet I, Eid AH, Hamam B. Knowledge, attitude and practices related to COVID-19 among young Lebanese population. BMC Public Health. 2021 Apr 6;21(1):653]

Second, their sampling is not matching this focus. Then there should be an age cap at 18 or maybe 21.

Answer: the aim of our study was to understand knowledge, attitudes, and behaviours of young Italians toward anti-SARS-CoV-2 vaccination. So we decided to focus our attention not only on adolescents, but also on older young people (over 18 years old). And according to our univariate and multivariate analysis, this was a reasonable choice, indicating a high level of willingness to get the vaccination in very young (under 18).

Third, for teenagers below 16, in most countries, parents or guardians are in charge of vaccination decisions. Thus, here the knowledge, attitude, behaviors of the teenagers <16 are less relevant. The authors should discuss this adequately, and potentially stratify their sample into an age-group that can take own decisions vs. those that cannot.

Answer: in Italy it is possible that people under 18 years old can refuse the vaccination, if they do not want to be vaccinated, so according to us this is an issue to be taken into account and teenagers’ behaviour is relevant.

We added in the Discussion section the following sentence:

“It is interesting to note that respondents in the age-group that can take own decisions showed a lower attitude to get the SARS-CoV-2 vaccination.”

The study does not mention ethics approval, especially since there are minors involved. This requires parental consent. I do not see how this was obtained in a survey. If parental consent is missing, this study severely violates ethics regulations. The authors need to clarify this

Answer: in the previous version the ethical approval was already mentioned as follows (lines295-297).

Institutional Review Board Statement: The study was conducted in accordance with the Declaration of Helsinki, and approved by the Institutional Review Board (Local Ethical Committee n.109/2020).

Moreover, the Consent to participate in the study was submitted as a supplementary file.

Reviewer 2 Report

thank you for asking me to review the ms. titled “Knowledge, attitude, behavior, toward SARS-CoV-2 vaccination in young Italians” by Kibi et al.

The paper aims at describing cross-sectional data on this topic to be used for improving vaccination campaigns among young population.

As presented, the paper has a number of flaws tough.

General issues

First of all, in the title and throughout the whole text authors repeated “SARS-CoV-2 vaccination”; while vaccines showed an ancillary effect against infection (also depending on circulating variants), vaccines are approved and licensed against severe disease and deaths, thus named “COVID-19 vaccine”.

I also suggest to avoid anecdotal sentence in the text. For instance, “On the 11th March 2020, WHO General Director Tedros Adhanom Ghebreyesus declared a pandemic of COVID-19”. Declaration of pandemic events are steered by the International Health Regulations (2005) Emergency Committee of the WHO; the WHO “simply” announces IHR Emergency Committee’s decisions.

Similarly, it is not clear why authors exclusively referred to Comirnaty vaccine. I understand it is the first to be authorized in Italy, but the study sample includes participants up to 30, a population targeted with other vaccines too. Moreover, the study was conducted in March 2021, when other vaccines were administered to this specific population group.

Regarding the study period, again, my doubts are about the validity of data collected almost 2 years ago, with a particular research work that has been outdated these days, especially in the light of the methodological questions raised below.

Methodological issues

More on the used survey should be said. It is not clear is they used a validated questionnaire, whether survey questions had been previously piloted, etc. Authors might want to include the survey as supplementary material.

Questions presented in Table 2 seem very unscientific: in some cases, multiple choices are no related one another and may have led respondents (young people with a scarce or no knowledge on biology/COVID-19/science in general) to random select the response.    

Study procedures should be better presented.

What is Skuola.net? A cursory presentation of it is needed. Is it involved in this kind of researches? If not, is its infrastructure reliable for public health research?

Ethical issues about the inclusion of people aged 17 years or younger should be clarified. In particular for the group aged 11-14!!!!!!!

From a statistical standpoint, I want to point out that at various points throughout the paper, the authors refer to their findings in terms of likelihood (probability). This is incorrect in light of the fact that they calculate odds ratios. They should describe their results in line with Persoskie A, Ferrer RA. A Most Odd Ratio: Interpreting and Describing Odds Ratios. Am J Prev Med. 2017 Feb;52(2):224-228. Briefly, since many outcomes are actually common in the sample (see percentages in Table 1), the risk that OR severely exceed the RR is high! Authors should change analysis towards weighted predicted probability.

Considering the above, results and discussion sections need revision aftee  methodological amendments of the whole ms.  

Author Response

Reviewer 2

thank you for asking me to review the ms. titled “Knowledge, attitude, behavior, toward SARS-CoV-2 vaccination in young Italians” by Kibi et al.The paper aims at describing cross-sectional data on this topic to be used for improving vaccination campaigns among young population. As presented, the paper has a number of flaws tough.

 General issues

First of all, in the title and throughout the whole text authors repeated “SARS-CoV-2 vaccination”; while vaccines showed an ancillary effect against infection (also depending on circulating variants), vaccines are approved and licensed against severe disease and deaths, thus named “COVID-19 vaccine”.

Answer: many thanks for this comment.

We replaced in the whole text SARS-CoV-19 with Covid-19

I also suggest to avoid anecdotal sentence in the text. For instance, “On the 11th March 2020, WHO General Director Tedros Adhanom Ghebreyesus declared a pandemic of COVID-19”. Declaration of pandemic events are steered by the International Health Regulations (2005) Emergency Committee of the WHO; the WHO “simply” announces IHR Emergency Committee’s decisions.

Answer: many thanks for this suggestion.

The first sentence is now as follows:

“COVID-19 had a huge impact on the world's population: social isolation due to public health measures, such as quarantine, resulted in discomfort that was especially reflected in schools, where distance learning caused students symptoms of stress, interaction problems, and lack of motivation to study [1]. “

Similarly, it is not clear why authors exclusively referred to Comirnaty vaccine. I understand it is the first to be authorized in Italy, but the study sample includes participants up to 30, a population targeted with other vaccines too. Moreover, the study was conducted in March 2021, when other vaccines were administered to this specific population group.

Answer: we agree with the reviewer. The following sentences were added in the revised version of the manuscript:

On 29 January 2021 the CHMP gave a positive opinion regarding the use of another COVID-19 vaccine, ChAdOx1-S, indicated for active immunisation to prevent COVID 19 caused by SARS-Co-V2, in individuals 18 years of age and older. Until now, ChAdOx1-S is not currently authorised for use in children.

Regarding the study period, again, my doubts are about the validity of data collected almost 2 years ago, with a particular research work that has been outdated these days, especially in the light of the methodological questions raised below.

 Methodological issues

More on the used survey should be said. It is not clear is they used a validated questionnaire, whether survey questions had been previously piloted, etc. Authors might want to include the survey as supplementary material.

Answer: we agree with the reviewer. We used a validated questionnaire, and we report details in the Methods section.

The following statement was added:

A pilot study was carried out to test the reliability of the questionnaire, giving clear indication of good psychometric properties (Cronbach’s alpha = 0.794).

Questions presented in Table 2 seem very unscientific: in some cases, multiple choices are no related one another and may have led respondents (young people with a scarce or no knowledge on biology/COVID-19/science in general) to random select the response.   

Answer: this is unavoidable in every survey. In the explanation of the study, we informed participants about the need to report the best answer they could.

Moreover, for most questions there was the possibility to answer “I don’t know”, “I have no idea”, “Non response”, “I cannot answer this question”, so we believe that the random selection of the response should have been mitigated.

  •  

Study procedures should be better presented.

What is Skuola.net? A cursory presentation of it is needed. Is it involved in this kind of researches? If not, is its infrastructure reliable for public health research?

Answer: many thanks for this comment. In the methods we give much details on Skuola.net.

We added the following sentence:

“…. an Italian website for information and insights for secondary school, high school, and university students, already involved in other surveys [18-21].

Ethical issues about the inclusion of people aged 17 years or younger should be clarified. In particular for the group aged 11-14!!!!!!!

Answer: in the previous version the ethical approval was already mentioned as follows (lines295-297).

Institutional Review Board Statement: The study was conducted in accordance with the Declaration of Helsinki, and approved by the Institutional Review Board (Local Ethical Committee n.109/2020).

Moreover, the Consent to participate in the study was already submitted as a supplementary file.

From a statistical standpoint, I want to point out that at various points throughout the paper, the authors refer to their findings in terms of likelihood (probability). This is incorrect in light of the fact that they calculate odds ratios. They should describe their results in line with Persoskie A, Ferrer RA. A Most Odd Ratio: Interpreting and Describing Odds Ratios. Am J Prev Med. 2017 Feb;52(2):224-228. Briefly, since many outcomes are actually common in the sample (see percentages in Table 1), the risk that OR severely exceed the RR is high! Authors should change analysis towards weighted predicted probability.

Considering the above, results and discussion sections need revision aftee  methodological amendments of the whole ms. 

Answer: we do not understand this issue since in the text we never indicated the words likelihood or probability.

In order to avoid misunderstanding we rephrased the following statements:

“women show higher odds of getting vaccinated compared to men”.

“In fact,  this study showed that individuals who are of legal age, have lower odds of getting COVID-19 vaccine (OR = 0.46, 95% CI, 0.40 – 0.54).”

Round 2

Reviewer 1 Report

I have read a previous version of the MS and see where it is improved. I believe that it would help the MS if the authors include some of the explanations provided in the reply to the editors more extensively in the MS. For example in the letter they state:

Answer: in Italy it is possible that people under 18 years old can refuse the vaccination, if they do not want to be vaccinated, so according to us this is an issue to be taken into account and teenagers’ behaviour is relevant.

We added in the Discussion section the following sentence:

“It is interesting to note that respondents in the age-group that can take own decisions showed a lower attitude to get the SARS-CoV-2 vaccination.”

The sentence included in the Discussion does not caption the complexity that is necessary here. International readers will not get this specific possibility in Italy. I would kindly ask the authors to provide adequate explanations: Which minors can refuse vaccinations (or medical treatment in general), which age group, etc.

I am still concerned about the Ethics. The reason why I claimed that no ethics information was provided is that I overlooked the statement, admittedly, my bad. But at the same time, it is common practice to describe how informed consent was obtained - and this information is still missing. How did guardians of minor provide informed consent? The authors should include a short section under Methods to describe how this was operationalized. Also, information should be provided which IRB assessed the proposal.

Author Response

Answer to reviewer 1 second report

I have read a previous version of the MS and see where it is improved. I believe that it would help the MS if the authors include some of the explanations provided in the reply to the editors more extensively in the MS. For example in the letter they state:

Answer: in Italy it is possible that people under 18 years old can refuse the vaccination, if they do not want to be vaccinated, so according to us this is an issue to be taken into account and teenagers’ behaviour is relevant.

We added in the Discussion section the following sentence:

“It is interesting to note that respondents in the age-group that can take own decisions showed a lower attitude to get the SARS-CoV-2 vaccination.”

The sentence included in the Discussion does not caption the complexity that is necessary here. International readers will not get this specific possibility in Italy. I would kindly ask the authors to provide adequate explanations: Which minors can refuse vaccinations (or medical treatment in general), which age group, etc.

Answer: we appreciate the comments of the reviewer, and the following statements were added.

In case of vaccination of a minor, applying the general rules established in international charters (i.e., the 1989 New York Convention on the rights of the child, the 1997 Strasbourg Convention on the rights of the child) from the age of 12 the minor must in any case be heard. This right also has to be applied to minors who have the ability to discern, in the procedures that concern them, in order to arrive at identifying the most suitable measure in accordance with the principle of the “ best interest of the child”.  (Baldini G. Vaccinazione anti Covid: quando genitori e figli minori non sono d’accordo (Anti Covid vaccination: when parents and minor children disagree. Altalex 22/06/2021. Available at https://www.altalex.com/documents/news/2021/06/22/vaccinazione-anti-covid-quando-genitori-e-figli-minori-non-sono-accordo). There is evidence that parents' decisions and attitudes are strictly related to the immunization status of adolescents (Cupertino V, Bozzola E, De Luca G, Del Giudice E, De Martino G, Cannataro P, Tozzi AE, Corsello G. The awareness and acceptance of anti-COVID 19 vaccination in adolescence. Ital J Pediatr. 2022 Dec 9;48(1):194.). However, in the case a minor would get the vaccination and one or both parents do not want, the jurisprudence orientation is that a judge can temporarily "suspend" the ability of the parent opposed to the vaccine. In this sense, the Italian National Bioethics Committee has stated the minor needs to be heard by pediatrician, and the will of the minor must prevail, "as it coincides with the best interest of his psycho-physical health and public health" (Italian Committee for Bioethics. Covid-19 Vaccines & Adolescents. Rome 31 July, 2021.)

I am still concerned about the Ethics. The reason why I claimed that no ethics information was provided is that I overlooked the statement, admittedly, my bad. But at the same time, it is common practice to describe how informed consent was obtained - and this information is still missing. How did guardians of minor provide informed consent? The authors should include a short section under Methods to describe how this was operationalized. Also, information should be provided which IRB assessed the proposal.

Answer: thanks for this comment. We provided the following paragraph:

Ethical issues

The study was conducted in accordance with the Declaration of Helsinki, and approved by the Institutional Review Board (Sapienza University of Rome – Policlinico Umberto I Ethical Committee n.109/2020). Informed consent was obtained using a ad hoc consent form in the presentation of the survey. If the case guardians of minor provide informed consent.

Reviewer 2 Report

Dear authors, 

Regarding your response «Answer: we do not understand this issue since in the text we never indicated the words likelihood or probability.» 

Please, carefully read the previous version of your paper, which reads “(women are more likely to get vaccinated than men) (OR = 1.22, IC95%, 1.06 – 1.40) (line 177)”, “are less likely to get COVID-19 vaccine (OR = 0.46, 95% CI, 0.40 – 0.54)” (line 122): in both case, ORs were clearly interpreted (misinterpreted!) as likelihood.

Please, consider that - grammatically speaking -  the use of the adverb “likely” is the same that use the noun “likelihood”!!!), but in a different phrase construction. 

Not just a question of using or not the word likelihood or likely. The problem is how authors interpreted their results

Authors continued that “In order to avoid misunderstanding we rephrased the following statements: “women show higher odds of getting vaccinated compared to men”, “In fact, this study showed that individuals who are of legal age, have lower odds of getting COVID-19 vaccine (OR = 0.46, 95% CI, 0.40 – 0.54).” besides the fact that this is not a matter of ‘misunderstanding’, but of correct interpretation of a statistical analysis that the same authors decided to use.

Likelihood – and those who/wat are more or less LIKELY – refers to finding the best distribution of the data given a particular value of some feature or some situation in the data. Odds is the chance of an event occurring against the event not occurring.

Hope this may help in your future research. 

Again, authors did not addressed my other comment on correct nomenclature of vaccine against COVID-19. In fact, in lines 91 and 95 it is possible to read “SARS-CoV-2 vaccine”

Presentation of multivariable analysis raises doubts. Since authors did not use a stepwise approach, it would be important to add data on statistical significance level of the selected explanatory variables included in the model.

Again, the variable “Knowledge score” is built on as a point-defined scale , this should not be included in the model as a continuous variable, but as an interval scale  Example: Ref 1 vs. 2 vs 3.

As said before, please avoid anecdotal sentence in the text: lines 220-222. Rephrase for clarity. 

Limitation section is poor. 

While (IMO) the fact “The questionnaire did not discriminate between different types of vaccines” is not a limitation (for instance, KAP studies on vaccination could never take into account different vaccine type and platform… think about flu vaccines), authors might want to include some aspect relating the sampling performed.

Overall, amendment in the presentation of text results are needed: number with three o two decimals, 95%IC instead of 95%CI, date presented as DD/MM/YEAR and MM/DD/YEAR (except of June 31st 30th, 2021) or the use of ordinal numbers (st, nd, rd, th).

Author Response

Answer to reviewer 2 second round

Dear authors, 

Regarding your response «Answer: we do not understand this issue since in the text we never indicated the words likelihood or probability.» 

Please, carefully read the previous version of your paper, which reads “(women are more likely to get vaccinated than men) (OR = 1.22, IC95%, 1.06 – 1.40) (line 177)”, “are less likely to get COVID-19 vaccine (OR = 0.46, 95% CI, 0.40 – 0.54)” (line 122): in both case, ORs were clearly interpreted (misinterpreted!) as likelihood.

Please, consider that - grammatically speaking -  the use of the adverb “likely” is the same that use the noun “likelihood”!!!), but in a different phrase construction. 

Not just a question of using or not the word likelihood or likely. The problem is how authors interpreted their results

Authors continued that “In order to avoid misunderstanding we rephrased the following statements: “women show higher odds of getting vaccinated compared to men”, “In fact, this study showed that individuals who are of legal age, have lower odds of getting COVID-19 vaccine (OR = 0.46, 95% CI, 0.40 – 0.54).” besides the fact that this is not a matter of ‘misunderstanding’, but of correct interpretation of a statistical analysis that the same authors decided to use.

Likelihood – and those who/wat are more or less LIKELY – refers to finding the best distribution of the data given a particular value of some feature or some situation in the data. Odds is the chance of an event occurring against the event not occurring.

Hope this may help in your future research. 

Answer: we really thank the reviewer for these comments and suggestions.

Again, authors did not addressed my other comment on correct nomenclature of vaccine against COVID-19. In fact, in lines 91 and 95 it is possible to read “SARS-CoV-2 vaccine”

Answer: corrections made

Presentation of multivariable analysis raises doubts. Since authors did not use a stepwise approach, it would be important to add data on statistical significance level of the selected explanatory variables included in the model.

Answer: In the Methods section the following sentence was added:

Univariate analyses were considered to test the inclusion of each explanatory variable in the final models. Effects with a P-value <0.20 were included in the multivariate analysis model.

Again, the variable “Knowledge score” is built on as a point-defined scale , this should not be included in the model as a continuous variable, but as an interval scale  Example: Ref 1 vs. 2 vs 3.

Answer: Thanks for this suggestion. According to this, modifications were made in the text

“increasing knowledge score was associated to higher odds of the willingness to be vaccinated (OR = 1.89, 3.10 and 3.61 for a score of 1, 2 and 3, respectively)”

and in the table 4

Knowledge score

0 (reference)

1

2

3

1

1.89 (1.45 – 2.48)

3.10 (2.40 – 3.99)

3.61 (2.76 – 4-71)

As said before, please avoid anecdotal sentence in the text: lines 220-222. Rephrase for clarity. 

Answer: rephrased as requested

Limitation section is poor. 

While (IMO) the fact “The questionnaire did not discriminate between different types of vaccines” is not a limitation (for instance, KAP studies on vaccination could never take into account different vaccine type and platform… think about flu vaccines), authors might want to include some aspect relating the sampling performed.

Answer: we modify the paragraph as follows:

First of all, a methodological difficulty concerns the truthfulness of answers, and risks of biases in population-based surveys, such as information and recall bias, cannot be excluded. Secondly, participants were only students who visited the “Skuola.net” web-site and a selection bias cannot be excluded. Since the recruitment method is based on voluntary participation, the sample may not be representative of the population. However previous surveys performed with skuola.net showed results similar to those carried out with sample-based surveys. Finally, in this survey family-socioeconomic status was not assessed, and it is possible that some factors such educational level of parents could have biased the results [30-32]

Overall, amendment in the presentation of text results are needed: number with three o two decimals, 95%IC instead of 95%CI, date presented as DD/MM/YEAR and MM/DD/YEAR (except of June 31st 30th, 2021) or the use of ordinal numbers (st, nd, rd, th).

Answer: modified as requested

Round 3

Reviewer 1 Report

Thanks for submitting a revision and for addressing my concerns. I believe they are adequately dealt with.

I would kindly ask the authors to have their last revisions thoroughly proof-read

For example in line 147 the sentence does not make sense.  I think it must read: "In case of minors, guardians provided informed consent."

line 285: directly, not strictly

line 287: agree, not want

line 289: either "the pediatrician" or "pediatricians"

line 290 in the quote: please add "her", as the quote only refers to male minors: his (her)

line 325: not difficulty, but "concern"

line 329: not excluded, but "ruled out"

etc.

Author Response

Reviewer 1 – third report

Thanks for submitting a revision and for addressing my concerns. I believe they are adequately dealt with.

I would kindly ask the authors to have their last revisions thoroughly proof-read

For example in line 147 the sentence does not make sense.  I think it must read: "In case of minors, guardians provided informed consent."

line 285: directly, not strictly

line 287: agree, not want

line 289: either "the pediatrician" or "pediatricians"

line 290 in the quote: please add "her", as the quote only refers to male minors: his (her)

line 325: not difficulty, but "concern"

line 329: not excluded, but "ruled out"

etc.

Answer: we checked all the items suggested by the reviewer, and corrected them.

Reviewer 2 Report

Accept 
